# Back to life: Techniques for developing high-quality 3D reconstructions of plants and animals from digitized specimens

Elizabeth G. Clark[1]*, Kelsey M. Jenkins[2], Craig R. Brodersen[3]

**1** Department of Environmental Science, Policy and Management, University of California Berkeley, Berkeley, California, United States of America, **2** Department of Earth and Planetary Sciences, Yale University, New Haven, Connecticut, United States of America, **3** School of the Environment, Yale University, New Haven, Connecticut, United States of America

* elizabethclark@berkeley.edu

**Data Availability Statement:** Data is available as part of the Supplemental Materials and from Figshare: https://doi.org/10.6084/m9.figshare.21266568.

## Abstract

Expanded use of 3D imaging in organismal biology and paleontology has substantially enhanced the ability to visualize and analyze specimens. These techniques have improved our understanding of the anatomy of many taxa, and the integration of downstream computational tools applied to 3D datasets have broadened the range of analyses that can be performed (e.g., finite element analyses, geometric morphometrics, biomechanical modeling, physical modeling using 3D printing). However, morphological analyses inevitably present challenges, particularly in fossil taxa where taphonomic or preservational artifacts distort and reduce the fidelity of the original morphology through shearing, compression, and disarticulation, for example. Here, we present a compilation of techniques to build high-quality 3D digital models of extant and fossil taxa from 3D imaging data using freely available software for students and educators. Our case studies and associated step-by-step supplementary tutorials present instructions for working with reconstructions of plants and animals to directly address and resolve common issues with 3D imaging data. The strategies demonstrated here optimize scientific accuracy and computational efficiency and can be applied to a broad range of taxa.

## Introduction

The use of three-dimensional (3D) visualization techniques have been used in biology and paleontology for over a century [1, 2]. Digital 3D visualization techniques such as computed tomography (CT) have become widespread within the last decade [e.g., 3–5]. One of the primary advantages of digital 3D methods is their non-destructive nature that allows the user to virtually dissect a sample into thousands of virtual serial sections, which can then be viewed using computer visualization software to explore the sample beyond the traditional orthogonal planes available with light microscopy. CT imaging also permits the visualization of internal and external anatomical features as they are arranged *in situ*, which can be useful for

**Funding:** The author(s) received no specific funding for this work.

**Competing interests:** The authors have declared that no competing interests exist.

interpreting function and behavior [e.g., 6–9]. CT imaging can be especially useful in the visualization of fossil taxa that may be embedded within a rocky matrix [e.g., 10–12].

Using 3D image processing software, CT image data or data from serial sectioning can be used to create a 3D reconstruction of a fossil or extant specimen. This is done by manually or automating the selection of **voxels** (three-dimensional pixels) that designate the specimen or region of interest in the 3D image. The shape of the selected voxels can then be used to generate a **mesh** (a digital polyhedron) comprised of a series of triangular **faces** made of points (**vertices**) connected by line segments (**edges**) formed about the surface of the selected voxels (Fig 1). This mesh can be exported from the 3D image processing software for subsequent analyses.

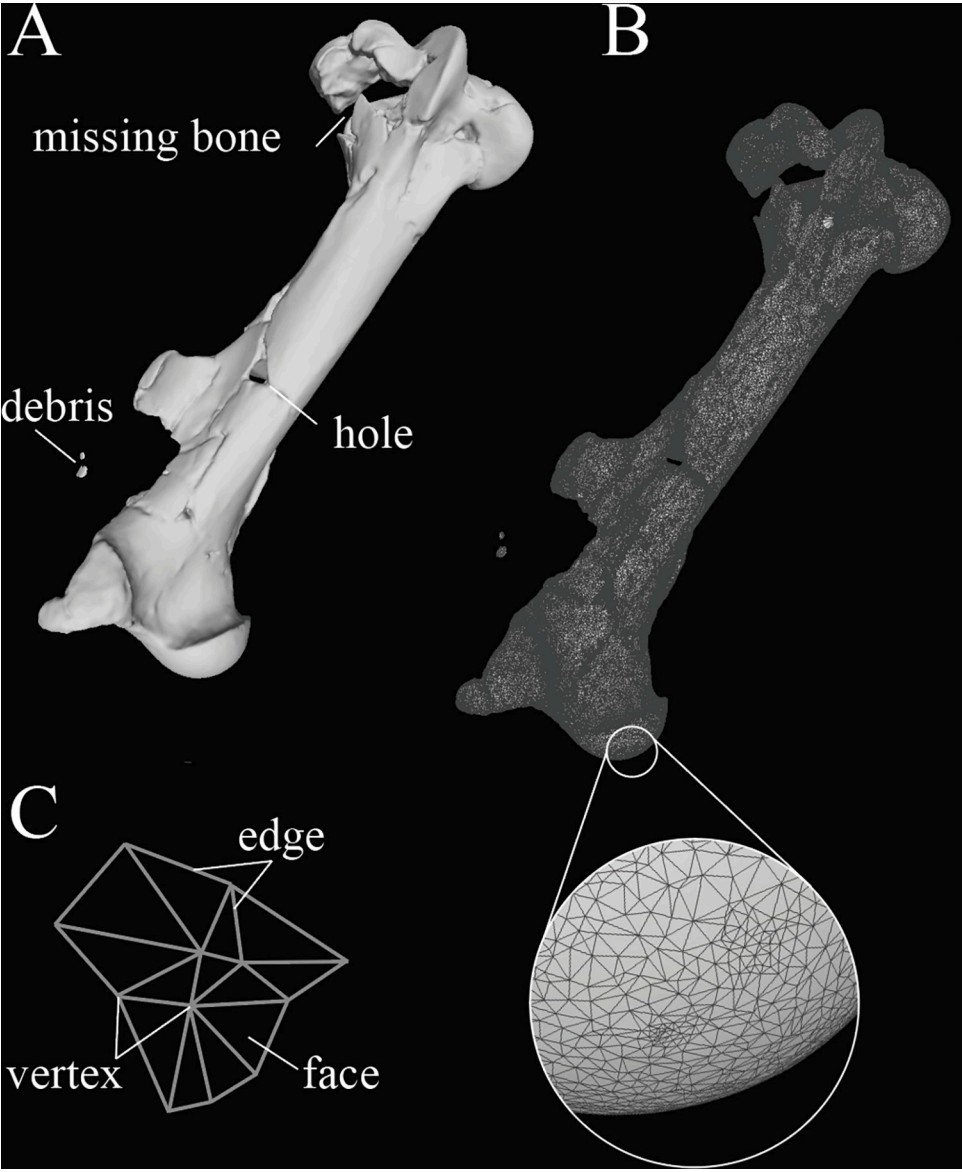

**Fig 1.** (A) 3D reconstruction of a left femur from *Equus* sp. (UF 234402) downloaded from morphosource.org and visualized using Meshlab [20]. Specimen demonstrates several preservational issues commonly encountered while working with 3D imaging data (labeled). (B) View of the mesh components that comprise the 3D reconstruction, which includes (C) edges, vertices and faces.

In studies of functional morphology, 3D reconstructions integrating or informed by CT imaging can be used in downstream analyses such as geometric morphometrics, finite element analyses, biomechanical analyses, fluid dynamics simulations, physical modeling using 3D printing, and more [8, 13–19].

However, digital volumes of whole specimens or even isolated anatomical structures directly extracted from 3D imaging data are generally suboptimal for surface visualization and downstream analyses due to structural inaccuracies which can be acquired during the processes of preservation, sample preparation, and mesh creation. During preservation, samples can incur structural damages which 3D digital visualization alone does not rectify. Sample preparation can also integrate structural inaccuracies in 3D visualization. The generation of a 3D volume with CT scanning relies on a density contrast between the sample or anatomical features of interest and their surroundings; objects embedded within material of similar densities, for instance, can have low contrast and result in a 3D volume of low quality. Many meshes generated from 3D visualization software can be extremely large and require simplification due to computational limitations, which can remove detail. Creating a likeness of a specimen by sculpting a 3D model from a simpler shape (such as a sphere or cube) via manual translation of vertices, edges, and faces (i.e., box modeling) has become a common way to circumvent these issues [17, 18, 21–24]. Ideally, however, 3D reconstructions would directly incorporate as much data from the original digitized specimen as possible. Field-specific techniques have been developed to reconstruct damaged or missing anatomical components based on known morphologies of similar, closely related fossil taxa with few taphonomic artifacts [25–38]. However, every digitized specimen comes with unique anatomy and its own set of structural issues.

The goal of this paper is to provide a set of tools for developing customizable workflows to build an optimal 3D digital reconstruction for specimens across taxonomic boundaries. Using 3D editing software that is freely available for students and for educational and research purposes, we present a compilation of strategies for repairing common taphonomic and preservational artifacts in 3D meshes of digitized extant and fossil specimens that can be applied broadly across plants and animals in a series of step-by-step tutorials (S1 File). These workflows are designed to be applied to meshes extracted from digitized specimen data and improve the fidelity of 3D digital reconstructions of a wide range of specimens. We include three case studies using plant, vertebrate, and invertebrate specimens, respectively, which include detailed tutorials for user learning and practice (S1 File). Elements of the workflows presented here have been employed to build 3D reconstructions of specimens for several studies [e.g., 5, 10, 12].

A 3D reconstruction is, by definition, a representation of a physical specimen. While repair and touch-ups of specimens may obscure original data, a carefully constructed model can often better convey the organism as it was *in vivo* and enable computationally intensive downstream analyses. This permits the application of digital technologies to study organisms in new ways, opening doors to illuminate previously unanswerable questions in organismal and evolutionary biology as well as enabling digital data dissemination and promoting accessibility.

## Materials and methods

Common issues in digitized specimens include the presence of external debris, shearing and compression, disarticulated, broken or missing elements, irregular surfaces and rough edges, large file size, and data storage. Here, we present a collection of strategies for addressing these and demonstrate their use in a set of user-friendly tutorials (S1 File) using the open-source software Meshlab [20] and Autodesk Maya and Fusion 360 (freely available under a student/

educator license). These tutorials are designed for users of all experience levels and taxonomic interests, although digitally correcting structural problems outside these examples will require a detailed understanding of an intact specimen to guide the process. The collection of techniques described in this article are available in the supplementary information (S1 File) and through protocols.io (dx.doi.org/10.17504/protocols.io.kqdg391deg25/v1).

## Removing external debris

The presence of external debris can hinder the digital visualization and preparation of both fossil and extant specimens. 3D imaging requires that the specimen is immobilized for the duration of the scan. This typically includes mounting the specimen in foam, plastic, or another low-density material. While it is a necessary step in data acquisition, the material used for specimen containment can be captured in the reconstructed image files, particularly when conducting extremely high-resolution imaging, such as synchrotron-based tomography. Furthermore, extracted volumes from 3D images of fossil specimens often retain portions of rock matrix. The removal of external debris can be incredibly time-consuming when performed manually; however, aspects of this process can be automated. Because digitized fossil elements are often one or several connected meshes, removing isolated mesh elements smaller than the primary mesh of the specimen can be an effective way to remove extraneous elements in bulk (demonstrated in Case Study 3). For noise represented by larger meshes, faces and vertices can be manually selected and deleted (demonstrated in Case Study 3). Extraneous faces connected to the desired object can also be smoothed out or removed (demonstrated in Case Study 2 and 3).

## Patching holes

At times, holes in the polygon mesh representing the surface structure resulting from taphonomic processes are present in fossils, or even in modern specimens that have missing or broken elements pre- or post-mortem. Following the rearticulation of disassociated elements (as demonstrated in Case Study 3), a secondary mesh (i.e., a **patch**) can be sized and sculpted in a modeling software to repair holes in the digital specimen (demonstrated in Case Study 2). Depending on subsequent analyses taking place using the specimen, the user may need to combine the fossil mesh and patch by shrinkwrapping by digitally creating a new simplified mesh using the position of the vertices at the surface of the original object. This process is outlined in Case Study 2.

## Shearing and compression

Diagenetic processes can result in shearing or compression of fossil specimens [e.g., 39, 40]. However, in certain cases, planes of symmetry can be inferred to provide guidance for reconstructing damaged areas. Deshearing or decompression of 3D object files can be performed by editing selected faces, vertices and/or edges via translation or scaling. This approach can allow the user to select a specific volume of tissue that is broken or compressed and then manipulate those areas such that they can be restored to match the surrounding unaffected tissue. The appropriate degree of deshearing or decompressing to be performed on a 3D object file can be difficult to estimate. For many taxa, the assumption of bilaterality is appropriate, and deshearing can be carried out over an observed or calculated plane of symmetry [37]. In other cases, an examination of multiple specimens preserved in different orientations can provide guidance regarding the degree of deshearing or decompression to be applied. In either case, prior knowledge of the taxa or group is necessary to inform these reconstructions, and any editing

performed must be justified and detailed through accessible documentation. Deshearing and decompression can be performed using the mesh sculpting tools described in Case Study 2.

## Disarticulated elements

Particularly with fossils, single vertebrate bones, arthropod segments, or other single-unit organismal components *in vivo* may often be broken or disarticulated. Individual elements can be exported separately from the reconstructed data in the original 3D visualization software (e.g. Avizo, VGStudio) or in a mesh editing software (e.g., Meshlab). These isolated elements can then be positioned in a program for modeling or rigging (e.g., Autodesk Maya). The proper positioning can be inferred from modern analogues, anatomical inference, examination of other specimens, or a symmetrical element can be mirrored if a plane of symmetry exists in the organism. Rearticulation of elements is demonstrated in Case Study 3.

## Missing features

If bilaterality can be assumed, missing elements can be created from a symmetrical element. The element can be isolated in a mesh editing software (e.g., Meshlab) and copied, reflected and positioned in a modeling/rigging program (e.g., Autodesk Maya). This method is outlined in Case Study 3.

## Irregular surfaces

Many 3D meshes are extracted from 3D imaging software with uninformative internal structures and irregular surfaces, rendering the files large and poorly optimized for surface visualization. Additionally, the digitization process of certain specimens or anatomical features may inaccurately interpret and present a solid object as a series of floating, isolated faces or vertices (e.g., the left crab claw in Case Study 3). Individual faces and vertices can be manually removed in a mesh editing software (Case Study 3). They can also be softened by translating individual or groups of faces, edges, or vertices in a modeling program (Case Study 2). Prominent edges from lower-resolution polygon meshes can be smoothed in a mesh editing software using a mesh subdivision tool or a smoothing feature; however, both of these techniques have limitations. Mesh subdivision tools increase file size, while smoothing tools can reduce the quality of features of interest. This is detailed in Case Study 1.

These issues can be improved through application of a shrinkwrap, which is described in all three case studies. Shrinkwrapping an object can solve these issues by creating a simplified 3D mesh of the object's surface shape. A shrinkwrap can be used to further combine multiple elements if desired. We demonstrate two methods by which to undertake shrinkwrapping: the first method utilizes Meshlab in Case Study 1 and 3, and the second utilizes Autodesk Fusion 360 in Case Study 2.

## Large file size

Exported meshes from 3D image processing software can often be large and computationally intensive to process. There are several methods to reduce total file size in Meshlab and Autodesk Fusion 360, such as applying tools that remove excess components or mesh simplification (described in Case Study 1 and 2). If only the surface structure is desired, another approach is to demarcate internal faces and vertices and select those for removal (described in Case Study 1). Many of these tools reduce the quality of the 3D model, so the approach and extent of file reduction should be appropriate for the desired image quality.

## Data storage

Best practices for scientific investigations that generate 3D imaging data include making files openly accessible in perpetuity for other investigators. However, these datasets can be quite large (>10gb), making long-term file storage challenging. Several online data repositories are available for depositing and accessing 3D imaging data such as micro-CT image datasets and 3D object files. These include MorphoSource (morphosource.org), figshare (figshare.org), and Dryad (datadryad.org). 3D object files can be concurrently stored in an open 3D object repository such as Makerbot's Thingiverse, which has dedicated options for 3D printing (thingiverse.com), or Sketchfab, which has special options for creating virtual and augmented reality experiences (sketchfab.com). The unedited and edited 3D object files, the detailed workflow used to generate the edited files, as well as the original 3D imaging data should be made accessible in perpetuity [41–43].

# Results

A variety of combined methods can be used to create 3D digital models that are suitable for a diverse array of uses, from 3D printing for educational purposes to biomechanical modeling for locomotory analyses. We present three case studies (one fossil plant, one vertebrate, and one arthropod), each with different post-processing needs, to demonstrate example workflows to improve 3D models of digitized specimens. Step-by-step guidelines for performing the workflows used in these case studies are available in the supplementary information, along with the associated 3D object files (S1 File). The 3D object files used for model building were segmented and exported from micro-CT data using commercially available visualization and processing software (e.g. VG studio, Avizo) or obtained via a digital repository (e.g. Morphosource.org). These exported 3D object files of digitized specimens all required post-processing prior to use in downstream analyses. While segmentation software is often available at a cost, with some exceptions (e.g., SPIERS, 3D Slicer, Dragonfly), many of the post-processing software packages are freely available, particularly for student or educational use. These case studies highlight and address common problems that occur with 3D reconstructions of extant and fossil specimens. The amount and types of post-processing an individual 3D model requires is highly dependent on the specimen and intended use. Accurate reconstructions are facilitated by previous knowledge of the structure of the specimen or taxa, an understanding of how diagenesis, preservation or segmentation may have contributed to artifacts, an analysis of multiple specimens, and sensitivity test for determining variation caused by the digital workflow used. Thorough and accessible documentation of the steps taken to produce the 3D reconstruction should be provided.

## Plant case study

Many fossil and extant plants possess unique anatomical features with distinctive shapes or unusual planes of symmetry. This can create challenges for generating 3D reconstructions, as certain processing tools are optimized for elongate structures (i.e., bones) or objects with bilateral symmetry. Here, we present a broadly applicable workflow for creating 3D digital reconstructions through a case-study using *Petriellaea triangulata*, a cupule of a fossil plant from the Early Cretaceous of Inner Mongolia [12] (S1 File). This specimen was segmented using a digital thresholding tool and extracted as a 3D object file from Avizo. Due to taphonomic affects and a low density contrast in the micro-CT scan that necessitated manual segmentation, the 3D object file representing the cupule contains preservational artifacts including ridges, rough surfaces, and holes (Fig 2A). These issues were rectified by performing a protocol to shrink-wrap the specimen. Shrinkwrapping entails using the vertices that comprise the surface of the mesh to create a new mesh layer that simplifies the shape of the original 3D object. This fills

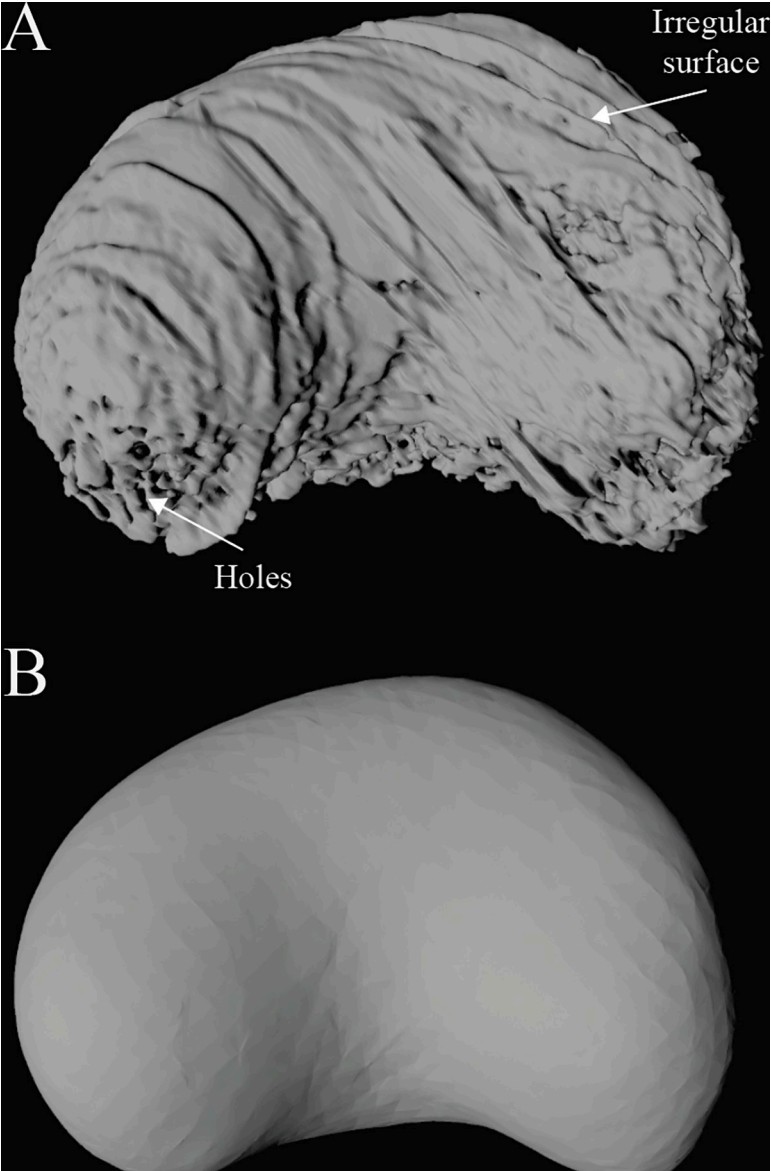

**Fig 2.** 3D reconstruction of *Petriellaea triangulata* [12], before (A) and after (B) post-processing. A. 3D mesh [12] exported directly from Avizo version 9.2.0 (Thermo Fisher Scientific, Massachusetts, USA). Object file exhibits many of the common issues described for fossil taxa, such as holes and irregular surfaces, which represent artefacts from preservation, low density contrast in the micro-CT image, and manual segmentation. B. 3D mesh file post-processed in Meshlab. Images generated using Autodesk Maya.

holes and smooths uneven surfaces while minimizing the loss of anatomical fidelity. This process involves removing internal and external faces and internal vertices in Meshlab and applying a surface reconstruction tool to build a watertight mesh averaging the position of the remaining vertices (Fig 2B).

## Vertebrate case study

Broken or missing elements are a common issue among specimens, particularly fossils. Even in specimens in which most elements are preserved *in situ* or are easily rearticulated, small

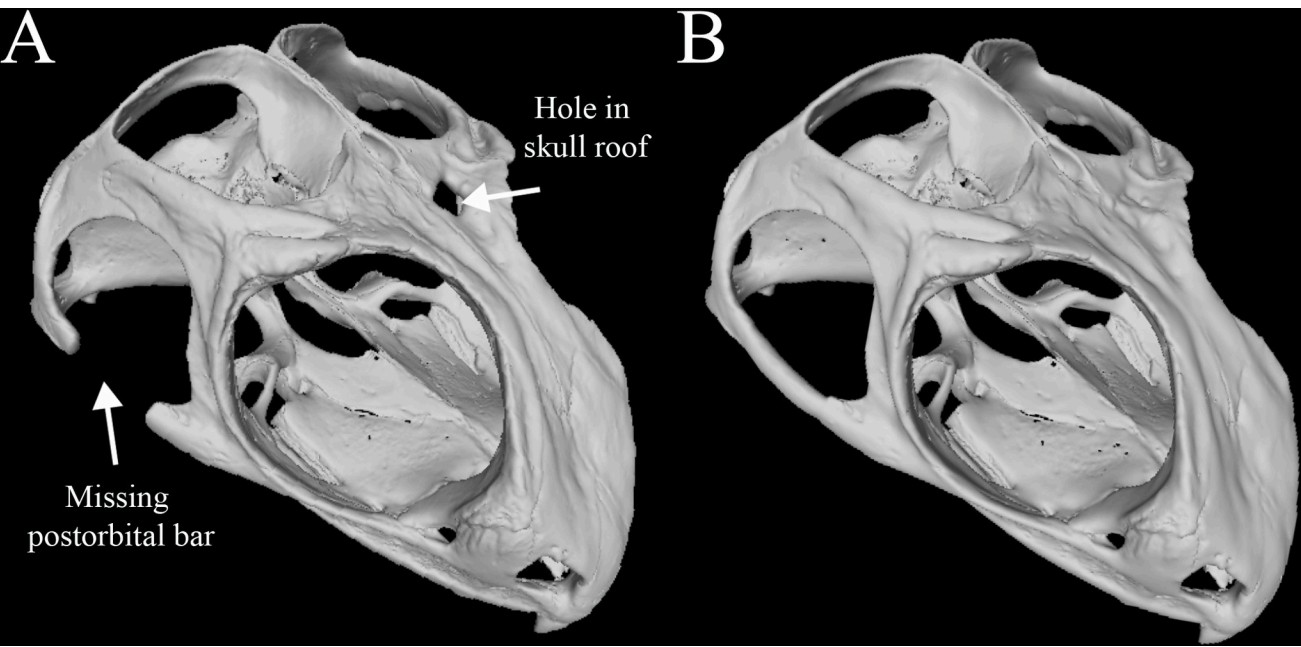

**Fig 3.** 3D reconstruction of *Sphenodon punctatus* (UF 11978, Florida Museum of Natural History) before (A) and after (B) post-processing. 3D mesh was downloaded directly from Morphosource.org. (A) Mesh was artificially damaged to create a hole in the roof of the skull and a missing lower temporal bar. B. 3D mesh file post-processed in Autodesk Maya and Autodesk Fusion 360 with patched hole and reconstructed temporal bar. Images generated using Meshlab.

holes and missing features may still exist. In this example, a mesh of a skull of *Sphenodon punctatus* (UF 11978, Florida Museum of Natural History, downloaded from Morphosource.org) was artificially damaged to replicate common breaks seen in extant and fossil specimens (Fig 3). A hole was created in the left frontal, and the right postorbital bar was removed to replicate common breaks seen in extant and fossil specimens.

To patch the smaller hole in the frontal, a mesh cube was created in Autodesk Maya and then scaled to the approximate size of the hole. The edges of the cube were buried within the margins of the mesh of the skull, and the planar surface of the cube was sculpted using translational and rotational tools. Once the cube was sculpted to match the surrounding areas of the skull, the two meshes were combined. To repair the right postorbital bar, a mesh cylinder was scaled and shaped to match the postorbital bar on the left side of the skull and was combined with the skull mesh. While combining the mesh in Autodesk Maya does fuse the patch with the rest of the model, they are still functionally two independent meshes that may or may not complicate downstream analyses. Depending on the use of the model, no further steps may be required. However, some subsequent analyses require a "water tight" digital model, opposed to the combined meshes that are present in the skull of *Sphenodon punctatus* at this state. A shrinkwrap was applied in Autodesk Fusion 360 at this stage to create a subsequent surface model of the skull and the two patches. This creates a single model without internal faces caused by the two patches. Further simplification of the model was also performed by reducing the face count in order to reduce the file size.

### Invertebrate case study

3D digital imaging of fossils can capture portions of rock matrix (i.e., debris) along with the fossilized organism. This material can often be easily removed in 3D visualization software

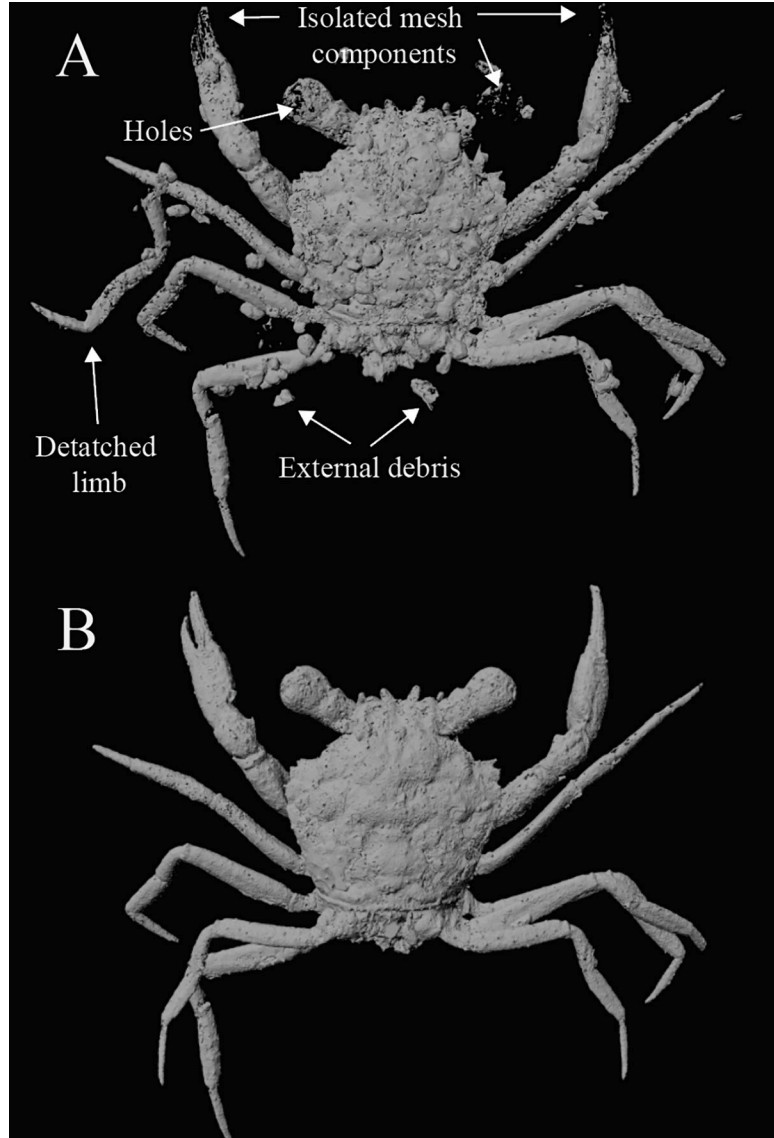

**Fig 4.** 3D reconstruction of *Cretapsara athanata* [5] before (A) and after (B) post-processing. (A) 3D mesh [5] exported directly from VG Studio MAX version 3.0. Object file exhibits many of the common issues described for fossil taxa, including the presence of external debris, disarticulated elements, isolated or missing mesh elements, holes, and irregular surfaces. (B) 3D mesh file post-processed in Meshlab and Autodesk Maya. Images generated using Autodesk Maya.

during segmentation prior to 3D mesh creation. However, extraneous material may be of a similar density to the specimen itself or directly affixed to the digitized organism, and some additional processing may be necessary. A digitized fossil crab of *Cretapsara athanata* preserved in amber [5] represents the most complete fossil crab ever discovered to date. However, it still exhibits many common preservational issues, including the presence of external debris, disarticulated and missing elements, holes, and rough edges and irregular surfaces (Fig 4A). After the crab was manually segmented and exported as a 3D mesh from VG Studio, post-processing was conducted in Meshlab and Autodesk Maya. The removal of small external debris was automated in Meshlab because the 3D mesh of the crab has a much larger number of

connected faces than the smaller, isolated elements desired for removal. Larger pieces of external debris and debris affixed to the specimen were removed manually. The debris removal process creates large gaps in the mesh that were further repaired in Meshlab. To do this, a copy of the mesh was shrinkwrapped by removing the internal faces and vertices, converting the mesh to a point cloud, and applying a surface reconstruction tool [44]. Combining the shrink-wrapped mesh with a copy of the original mesh was performed to fill holes and smooth rough surfaces.

The eye as well as the dactylus and propodus (i.e., the claw) were preserved and extracted as a series of floating, isolated faces and vertices. These features were each isolated and shrink-wrapped separately due to relatively lower density of connected faces in these areas compared to the rest of the crab mesh. The left fifth leg was reattached by isolating the leg from the rest of the body in Meshlab, exporting both as separate object files, and repositioning the leg on the body in Maya. The left claw and the left eye were copied, reflected, and attached to the appropriate location on the right side.

## Discussion

Different kinds of models are used across the natural sciences to illustrate components of complex phenomena, such as physical robots created to examine animal locomotion [e.g., 45], fluid dynamic simulations to illustrate marine animal physiology [e.g., 46], material deformation models to analyze plate tectonics [e.g., 47], or atmospheric models designed to illuminate climate dynamics [e.g., 48]. These models are all designed with the intended goal of reducing the complexity of the original system and to aid in analysis and subsequent interpretation. Similarly, simplified 3D models of organisms are designed to focus on specific anatomical features to illuminate their structure or function. By providing a compilation of techniques that address common issues in working with digitized biological and paleontological specimens, we address the first steps in undertaking more complex analyses that aid in understanding functional morphology. By removing extraneous noise, and adding detail that better represents true morphology, these refined models are capable of detecting morphological signals that, when subjected to downstream analyses, inform our understanding of these organisms and their physiology. As such, simplified models have become more commonplace in studies of functional morphology, homology, and paleoecology. Despite the simplified nature of these models, the interpretations made from them are crucial for understanding evolutionary phenomena over deep time.

The vast majority of the workflows available for generating specimen reconstructions in 3D focus on vertebrates. A primary intention of this paper is to expand the narrative to demonstrate a range of issues across different taxa, discuss the applicability of different software to address these, and highlight the suite of downstream analyses that could be performed. As such, we expressly provide a toolkit that would be useful for individuals interested in any group (Fig 5). The tutorials are written in such a way as to be accessible to anyone, from student researchers to experts looking to develop new skills, and for those interested in a range of analytical tools and taxa.

However, the choice of techniques and the degree to which these approaches are put into effect determine the resulting 3D model and can impact the results of the downstream analyses. Thus, aspects of the original specimen to include, simplify, or infer during the creation of a 3D model must be suitable for the type of analysis in question. For example, simplification of a digitized fossil is typically carried out prior to undertaking computational fluid dynamics simulations to reduce noise and facilitate computation [e.g., 17, 18, 46]. Similarly, retrodeformation, translation, and the repair of broken elements may be appropriate to simulate *in vivo*

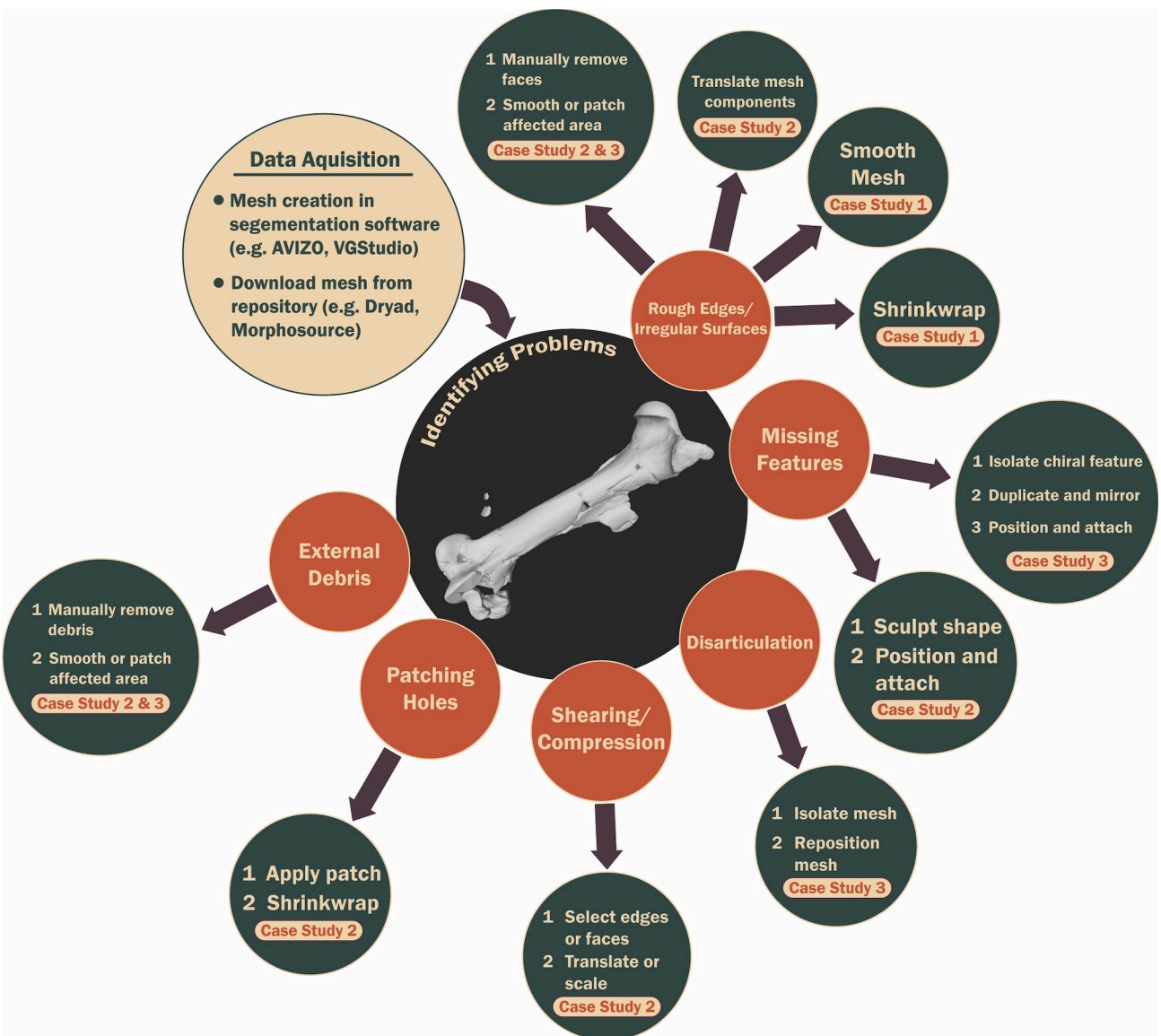

**Fig 5. Summary workflow to address common issues with 3D imaging data.** Step-by-step guidelines with examples and associated 3D object files can be found in the supplementary information (S1 File).

anatomy for biomechanical analyses. Sculpting a missing appendage into a 3D model of a fossil organism based on its nearest ancestor might not contain enough information to be appropriate for geometric morphometrics, for instance, but may be appropriate for building a virtual reality experience for scientific outreach. It is crucial that the kind of model postprocessing used is appropriate for the downstream analysis in question. If application of certain mesh editing techniques are deemed necessary for the performance of downstream analyses, it is strongly recommended that sensitivity analyses be performed (i.e., calculating the impact variation in the technique applied would have on the results of the analysis) [49].

Furthermore, steps used to create a 3D model must be transparent, justified, and accessibly detailed in all scientific publications in which they are used. The CT scan files, the unedited raw object files, the edited 3D models, and details regarding the editing performed should be

made accessible in a repository for long-term availability, and data from multiple specimens should be used and provided where possible. This collection of software and techniques that span taxonomic groups provides a step towards the wider accessibility and collective exchange of post-processing strategies and technology in the study of extant and fossil organisms. Resources are mentioned throughout the text that are available for additional information on certain techniques or taxon-specific approaches. We hope that this contribution can serve as part of an ongoing conversation to expand the network of solutions for improving 3D data processing and digital specimen reconstructions as well as their accessibility.

## Conclusion

3D imaging tools and processing software are becoming more accessible for researchers. However, many issues can limit the usefulness of these 3D reconstructions for different analyses. Here, we provide the workflows used to repair 3D meshes of specimens in a range of taxonomic groups extracted from micro-CT imaging data. These 3D models can be implemented for downstream uses such as morphological descriptions, geometric morphometrics, biomechanical modeling, 3D printing, and building creative educational tools. These approaches have been applied to a wide range of taxa including and beyond the three case studies presented here. We hope that the presentation of our case studies and supplementary tutorials enables the broader integration of high-quality 3D reconstructions in biological and paleontological analyses.

## Supporting information

**S1 File. Instructions to access tutorials and 3D object files.**
(DOCX)

## Acknowledgments

We are grateful to the members of the Brodersen Lab, the Briggs Lab, the Gauthier Lab and the Bhullar Lab (Yale University), the Structure & Motion Laboratory (Royal Veterinary College) and the Almeida Lab (UC Berkeley) for assistance and valuable discussion.

## Author Contributions

**Conceptualization:** Elizabeth G. Clark, Kelsey M. Jenkins, Craig R. Brodersen.

**Data curation:** Elizabeth G. Clark, Kelsey M. Jenkins.

**Investigation:** Elizabeth G. Clark, Kelsey M. Jenkins, Craig R. Brodersen.

**Methodology:** Elizabeth G. Clark, Kelsey M. Jenkins.

**Project administration:** Elizabeth G. Clark, Craig R. Brodersen.

**Resources:** Craig R. Brodersen.

**Software:** Craig R. Brodersen.

**Supervision:** Craig R. Brodersen.

**Validation:** Kelsey M. Jenkins.

**Visualization:** Elizabeth G. Clark, Kelsey M. Jenkins, Craig R. Brodersen.

**Writing – original draft:** Elizabeth G. Clark, Kelsey M. Jenkins, Craig R. Brodersen.

**Writing – review & editing:** Elizabeth G. Clark, Kelsey M. Jenkins, Craig R. Brodersen.

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
