## [Decision Letter · Decision Letter 0]

21 Nov 2022

PONE-D-22-27354Back to life: Techniques for developing high-quality 3D reconstructions of plants and animals from digitized specimensPLOS ONE

Dear Dr. Clark,

Thank you for submitting your manuscript to PLOS ONE. After careful consideration, we feel that it has merit but does not fully meet PLOS ONE’s publication criteria as it currently stands. Therefore, we invite you to submit a revised version of the manuscript that addresses the points raised during the review process.

You will see some comments the reviewers left, specially the first reviewer. I kindly ask you please to consider her observations and recommendations to improve the manuscript before publication. The comments of the first reviewer are attached in a .pdf file to this email. 

We look forward to receiving your revised manuscript.

Kind regards,

Judith Pardo-Pérez, Ph.D

Academic Editor

PLOS ONE

Journal Requirements:

Reviewers' comments:

Reviewer's Responses to Questions

**Comments to the Author**

1. Does the manuscript report a protocol which is of utility to the research community and adds value to the published literature?

Reviewer #1: Yes

Reviewer #2: Yes

2. Has the protocol been described in sufficient detail?

To answer this question, please click the link to protocols.io in the Materials and Methods section of the manuscript (if a link has been provided) or consult the step-by-step protocol in the Supporting Information files.

The step-by-step protocol should contain sufficient detail for another researcher to be able to reproduce all experiments and analyses.

Reviewer #1: Yes

Reviewer #2: Yes

3. Does the protocol describe a validated method?

Reviewer #1: Yes

Reviewer #2: Yes

4. If the manuscript contains new data, have the authors made this data fully available?

Reviewer #1: Yes

Reviewer #2: Yes

**5. Is the article presented in an intelligible fashion and written in standard English?**

Reviewer #1: Yes

Reviewer #2: Yes

6. Review Comments to the Author

Reviewer #1: The manuscript "Back to life: Techniques for developing high-quality 3D reconstructions of plants and

animals from digitized specimens" shows the work that E.G. Clark and colleagues have carried out to develop helpful protocols to create more realistic and useful 3D models to work with, specially when they present numerous artifacts. I highly appreciate the written protocols and examples, as they are easy to follow and comprehensible. However, I have several concerns:

- I feel the authors should acknowledge more the previous works on 3D modelling, retrodeformation and retopology, specially when presenting each technique, as there are already published protocols on them. Together with this, I would suggest to highlight the differences of the presented workflows and techniques with the previous ones. "Why should we follow your workflows? Why are they different to the already published? Are they easier/more straightforward to follow? Why should I use Maya instead of Blender, which is open source?". These are only suggestions that could be addressed within the discussion.

- I see an important issue on the "subjectivity" of the final models, as there is a lot of user interference within the workflows. The user's expertise on these software willl greatly impact the final output, and sometimes the process implies an extreme modification of the original mesh (e.g. the Petriellaea case study). Some kind of sensitivity analysis should be suggested, which can help to evaluate the reliability of the techniques, but also the user interference.

- A section on "how and when to use these outputs" would be useful, as my feelings are that they couldn't be of utility when highly accurate models are needed (e.g. some FEA, geometric morphometrics, comparison works,...). Again, this is also related with the "subjectivity artifact" indicated above.

- This is more a personal thought/suggestion: not all people have access to a commercial software, so I would suggest also including a workflow on an open-source program (like Blender, which has similar features as Maya). I understand the timeliness of the manuscript, but for future works this could be a nice idea to implement.

I also have annotated a few more comments on the pdf. I would like to congratulate the authors for such an interesting work, I really enjoyed working on the case studies and learnt new features.

Verónica Díez Díaz

Reviewer #2: General comments

The manuscript by Clark et al., provides an incredibly useful overview and step-by-step guide for overcoming many common problems experienced when working with 3D data. It was very well written and bound to be a highly useful and well received paper by the community.

As someone who works with 3D data from both fossil and extant specimens I thought the paper was fantastic and I look forward to recommending this to my lab as a useful reference. I also think the quality of the tutorials in supplementary information were brilliant – and definitely of a standard that could be used for teaching.

I have no major concerns with the manuscript and thank the authors for producing such a high quality and valuable resource.

I have a couple of recommendations which are centred around ways to help signpost the key information in the paper for a reader who may use this as a quick reference guide rather than as a paper to be read from start to finish.

1. I think you could make a summary workflow figure that captures the key parts of the paper and signposts the reader to different parts of the text. E.g. If you want to remove external debris from a specimen go to this section in text and this section in supplementary.

2. I think the common problems you list in methods “Removing external debris” etc could also be listed in a highlights box so someone glancing at the paper can spot the issue they are looking for.

3. I think it might be worth emphasising the step-by-step guide in supplementary slightly more in the main text. People often overlook supplementary info especially if someone just jumped to a case study that looked particularly useful for them.

4. I think you could possibly add some annotations or colour to the figures

Then two more general points to consider. I don’t think the MS necessarily needs these but could be something to consider:

1. You could mention that these techniques could also be particularly valuable for 3D reconstructions made from serial preparations that are inherently more blocky and prone to artefacts than CT data. E.g. serial sectioning, grinding or peeling.

2. Especially in the palaeo field it is very common to want to 3D print fossils. I know you mention 3D printing in the text but it could possibly be highlighted slightly more.

7. PLOS authors have the option to publish the peer review history of their article (what does this mean?). If published, this will include your full peer review and any attached files.

Reviewer #1: **Yes: **Verónica Díez Díaz

Reviewer #2: No

---

## [Author Response · Author response to Decision Letter 0]

4 Jan 2023

PONE-D-22-27354 Response to Reviewer Comments

We are grateful to both reviewers and the editor for their prompt review of our paper, and for their constructive comments which have substantially improved the manuscript. We have made the vast majority of suggested edits, with only a few exceptions. Comments from the reviewers are numbered below, with our response inserted beneath each one. Text that has been added to the main document is in red. Line numbers refer to the final submitted .docx version of the manuscript (please note that line numbers may have shifted slightly in the PDF compilation for review). 

Editor’s Comments

Changes made: We have updated the title page to reflect the formatting in the Title, Author, Affiliations Formatting Guide (lines 1-23). We have amended the figures to adhere to the figure preparation guidelines and file requirements. We have edited the main text and supporting information to match the manuscript body formatting guidelines. 

Changes made: We added a number of citations and corrected several references as noted below. To our knowledge, none of these references have been retracted.

Reviewer #1

Reviewer 1 had several suggestions regarding the inclusion of additional resources, such as previous modeling work and open-source software. They also suggested clarification for when different techniques are appropriate to use and to stress the importance of sensitivity analyses. They provided quite a few additional papers to cite in our work which will certainly help connect the readers with other resources and case studies.

1. I feel the authors should acknowledge more the previous works on 3D modelling, retrodeformation and retopology, especially when presenting each technique, as there are already published protocols on them. Together with this, I would suggest to highlight the differences of the presented workflows and techniques with the previous ones. "Why should we follow your workflows? Why are they different to the already published? Are they easier/more straightforward to follow? Why should I use Maya instead of Blender, which is open source?". These are only suggestions that could be addressed within the discussion.

Changes made: We view the strength of our manuscript in the wide array of tools in the toolkit presented, the applicability to a broad range of taxa, and the accessibility of the tutorials. We hope that this paper provides a resource to continue growing the network of techniques available for this kind of data processing. We added a statement to this effect on line 505-514 and 547-551, and added the number of additional references that the reviewer suggested for the users to access for further information (see responses to reviewer suggestions #6, #8, and #10). 

2. I see an important issue on the "subjectivity" of the final models, as there is a lot of user interference within the workflows. The user's expertise on these software will greatly impact the final output, and sometimes the process implies an extreme modification of the original mesh (e.g. the Petriellaea case study). Some kind of sensitivity analysis should be suggested, which can help to evaluate the reliability of the techniques, but also the user interference.

Changes made: We added a note regarding the importance of sensitivity analyses to determine the impact on the changes made to the resulting downstream analyses (365-370, 532-539). 

3. A section on "how and when to use these outputs" would be useful, as my feelings are that they couldn't be of utility when highly accurate models are needed (e.g. some FEA, geometric morphometrics, comparison works,...). Again, this is also related with the "subjectivity artifact" indicated above.

Changes made: Since each specimen is unique, we wanted to refrain from generalizing as to how and when downstream analyses may be appropriate. However, we did add a statement regarding information necessary to facilitate accurate model building (lines 365-370), including prior knowledge and sensitivity tests (161-164, 532-539). We have also added a figure that represents a summary workflow (Fig 5). We also have added a number of resources for additional information on specific techniques and downstream analyses and a note regarding this on line 547-551. We also added a note that thorough documentation of the steps used to produce an edited 3D reconstruction must be provided (369-370). 

4. This is more a personal thought/suggestion: not all people have access to a commercial software, so I would suggest also including a workflow on an open-source program (like Blender, which has similar features as Maya). I understand the timeliness of the manuscript, but for future works this could be a nice idea to implement.

Changes made: Maya is available for free through a student/educator license- we added this note to line 161. Furthermore, we have added resources including workflows using open-source software per the suggestion of the reviewer (e.g., Herbst et al.).

5. Line 72: Do you mean retrodeformation?

Changes made: We have added text to clarify this (line 113). 

6. Line 79: I totally understand that it is currently a complex task, staying updated with all the publications on retrodeformation techniques, but I strongly suggest you to include some of the previous works on them while presenting your case studies and protocols below, as other people have already done similar things.

Besides those you already include here, you can take a look at Demuth et al. (2022), who also has a really nice introduction with more works on these topics:

Demuth, O. E., Benito, J., Tschopp, E., Lautenschlager, S., Mallison, H., Heeb, N., & Field, D. J. (2022). Topology-based three-dimensional reconstruction of delicate skeletal fossil remains and the quantification of their taphonomic deformation. Frontiers in Ecology and Evolution, 125.

Check also:

Herbst, E. C., Meade, L. E., Lautenschlager, S., Fioritti, N., & Scheyer, T. M. (2022). A toolbox for the retrodeformation and muscle reconstruction of fossil specimens in Blender. Royal Society Open Science, 9(8), 220519.

Changes made: We have added these two references in the manuscript (lines 122-123, 647-650, 673-675). 

7. One more thing you should highlight, is why your protocols are noteworthy or different from those already published, where is their ground-breaking nature, why should researchers follow them.

Changes made: See changes made in response to suggestion #1. 

8. Line 98: Or the other way around, how could we be sure that this reconstructed model is accurate and closer to reality? How can we get rid of preconceived ideas and that we are following an unbiased approach? Is there any way to carry on a sensitivity analysis, for example? See below my comment on the Petriellaea case study, and e.g. Bishop et al. (2021) on the importance of these type of analyses when working with retrodeformed models:

Bishop, P. J., Cuff, A. R., & Hutchinson, J. R. (2021). How to build a dinosaur: Musculoskeletal modeling and simulation of locomotor biomechanics in extinct animals. Paleobiology, 47(1), 1-38.

Changes made: We have added this citation (line 533-539, 587-589), more information about the Petriellea specimen (lines 380-385, 398-400), and added a note about the importance and a strong suggestion for the performance of sensitivity analyses where possible (line 365-370, 532-539).

9. Line 159: Here, I would consider important that you cite previous works which deal on the articulation of digital bones/skeletons, as there are several things to take into account, and even some guidelines to follow. I include here some of them:

- On the osteological neutral pose:

Stevens, K. A., & Parrish, J. M. (1999). Neck posture and feeding habits of two Jurassic sauropod dinosaurs. Science, 284(5415), 798-800.

Stevens, K. A., & Parrish, J. M. (2005). Neck posture, dentition, and feeding strategies in Jurassic sauropod dinosaurs. In Thunderlizards: the sauropodomorph dinosaurs. (pp. 212-232). Indianan University Press.

Stevens, K. A. (2013). The articulation of sauropod necks: methodology and mythology. PLoS One, 8(10), e78572. 

- On the neutral bone only posture:

Vidal, D., Mocho, P., Páramo, A., Sanz, J. L., & Ortega, F. (2020). Ontogenetic similarities between giraffe and sauropod neck osteological mobility. PLoS One, 15(1), e0227537.

- On the cartilaginous neutral pose:

Taylor, M. P., & Wedel, M. J. (2013). The effect of intervertebral cartilage on neutral posture and range of motion in the necks of sauropod dinosaurs. PLoS One, 8(10), e78214.

Taylor, M. P. (2014). Quantifying the effect of intervertebral cartilage on neutral posture in the necks of sauropod dinosaurs. PeerJ, 2, e712.

- On the protocol on how to assemble 3D skeletons in order to minimize preconceived notions:

Mallison, H. (2010). The digital Plateosaurus II: an assessment of the range of motion of the limbs and vertebral column and of previous reconstructions using a digital skeletal mount. Acta Palaeontologica Polonica, 55(3), 433-458.

Changes made: In this part of the text, we are referring to fragmented single bones, segments, or other structural units. We added clarification to this on line 252-253. 

10. Line 201: Cite e.g.:

Davies, T. G., Rahman, I. A., Lautenschlager, S., Cunningham, J. A., Asher, R. J., Barrett, P. M., ... & Donoghue, P. C. (2017). Open data and digital morphology. Proceedings of the Royal Society B: Biological Sciences, 284(1852), 20170194.

Falkingham, P. L., Bates, K. T., Avanzini, M., Bennett, M., Bordy, E. M., Breithaupt, B. H., ... & Belvedere, M. (2018). A standard protocol for documenting modern and fossil ichnological data. Palaeontology, 61(4), 469-480. 

Moore, J., Rountrey, A., & Scates Kettler H. (2022). 3D Data Creation to Curation: Community Standards for 3D Data Preservation. Association of College and Research Libraries.

Changes made: We have added these resources to the text (line 309, 639-641, 657-659, 707-709).

11. Line 231: This is a really extreme modification from the original output to the post-processed model. Is there any way to assess the accuracy of the new mesh and the reliability of the process? Maybe develop comparison analyses with other better-digitized specimens, as a sensitivity analysis of the method?

Changes made: We clarified that the artefacts present in this specimen mostly stem from the density contrast that necessitated manual segmentation in the 3D imaging file (lines 380-385, 398-400 ). We added this to the text, and also added that accurate reconstructions is facilitated by previous knowledge of the structure of the specimen or taxa, an understanding of how diagenesis, preservation or segmentation may have contributed to artifacts, an analysis of multiple specimens, and sensitivity test for determining variation caused by the digital reconstruction workflow used (line 161-164, 365-367, 532-539). 

12. Line 238: (strikethrough Shi et al. 2021 text)

Changes made: We have made the change suggested (line 377-379). 

13. Line 272: For small-to-medium holes there are more straightforward/easier methods which imply less user manipulation. As you say in the supplementary guidelines, sculpting is an individual process. But some analyses require accurate meshes, so an "artsy" process should be avoided as much as possible. How could we evaluate this user interference in this workflow?

Changes made: We have added text suggesting sensitivity analyses be performed to address the impact of variation in the reconstruction techniques chosen by the user (line 365-370, 532-539), and added resources to connect users with more information and resources throughout the text as suggested by the reviewer. 

14. Line 276: So, the perpendicular-to-the-skullroof surfaces of the cube are embeded/merged with the rest of the mesh? Or are they eliminated in the shwrinkwrapping workflow with Autodesk Fusion 360? Sorry, I don't have a license for this software and cannot work on this part of the workflow.

Changes made: The reviewer is correct in that shrinkwrapping does eliminate these surfaces. We have added some clarification to this aspect (Lines 431-433).

15. 277: Same as above, this time the circular surfaces of the cylinder, those connecting with the postorbital bar.

Changes made: See response to item #14 above.

Reviewer #2: 

Reviewer 2 provided a number of helpful suggestions, such as adding a summary workflow, emphasizing the supplementary tutorials, and adding more information about techniques such as serial sectioning and 3D printing.

1. I think you could make a summary workflow figure that captures the key parts of the paper and signposts the reader to different parts of the text. E.g. If you want to remove external debris from a specimen go to this section in text and this section in supplementary.

Changes made: We added a new figure to easily walk the reader through common issues with meshes and how to address them (Fig 5), summarizing much of the information presented in the Materials and Methods in a visual schematic.

2. I think the common problems you list in methods “Removing external debris” etc could also be listed in a highlights box so someone glancing at the paper can spot the issue they are looking for.

Changes made: See response to #1 above.

3. I think it might be worth emphasising the step-by-step guide in supplementary slightly more in the main text. People often overlook supplementary info especially if someone just jumped to a case study that looked particularly useful for them.

Changes made: We added references the step-by-step guide in the supplement to several places in the manuscript (line 46, 130-131, 143, 159, 318-320, 379, 516-518). 

4. I think you could possibly add some annotations or colour to the figures

Changes made: We added annotations to show the issues to which we refer and how they manifest in the specimens we provided in the case studies (Figures 2-4) and added a color summary figure (Fig 5).

5. You could mention that these techniques could also be particularly valuable for 3D reconstructions made from serial preparations that are inherently more blocky and prone to artefacts than CT data. E.g. serial sectioning, grinding or peeling.

Changes made: We have added a note about other techniques for tomography (line 53-55), and that data from serial sectioning can also be used to generate 3D models (line 66), as well as two references (lines 629-630, 748-749). 

6. Especially in the palaeo field it is very common to want to 3D print fossils. I know you mention 3D printing in the text but it could possibly be highlighted slightly more.

Changes made: We have highlighted 3D printing more, both as an educational tool and a tool for physical modeling (line 40, 86, 315, 564-565).

---

## [Decision Letter · Decision Letter 1]

1 Mar 2023

Back to life: Techniques for developing high-quality 3D reconstructions of plants and animals from digitized specimens

PONE-D-22-27354R1

Dear Dr. Clark,

We’re pleased to inform you that your manuscript has been judged scientifically suitable for publication and will be formally accepted for publication once it meets all outstanding technical requirements.

Kind regards,

Judith Pardo-Pérez, Ph.D

Academic Editor

PLOS ONE

Additional Editor Comments (optional):

Reviewers' comments:

Reviewer's Responses to Questions

**Comments to the Author**

1. Does the manuscript report a protocol which is of utility to the research community and adds value to the published literature?

Reviewer #1: Yes

Reviewer #2: Yes

2. Has the protocol been described in sufficient detail?

To answer this question, please click the link to protocols.io in the Materials and Methods section of the manuscript (if a link has been provided) or consult the step-by-step protocol in the Supporting Information files.

The step-by-step protocol should contain sufficient detail for another researcher to be able to reproduce all experiments and analyses.

Reviewer #1: Yes

Reviewer #2: Yes

3. Does the protocol describe a validated method?

Reviewer #1: Yes

Reviewer #2: Yes

4. If the manuscript contains new data, have the authors made this data fully available?

Reviewer #1: Yes

Reviewer #2: Yes

**5. Is the article presented in an intelligible fashion and written in standard English?**

Reviewer #1: Yes

Reviewer #2: Yes

6. Review Comments to the Author

Reviewer #1: I would like to congratulate the authors for the revision and how they have dealt with the reviewers' comments, as the work has improved considerably. Especially the inclusion of figure 5 seems to me a very smart idea, which explains in a very visual and intuitive way all the protocols and how to proceed.

For my part I can add nothing more, only that the authors include the reviewers in the acknowledgements in the final version of the manuscript. I hope to see this work published soon.

Verónica Díez Díaz

Reviewer #2: I have no further comments. I thank the authors for taking on board my early suggestions. I think the additional changes they have made have strengthened the manuscript.

7. PLOS authors have the option to publish the peer review history of their article (what does this mean?). If published, this will include your full peer review and any attached files.

Reviewer #1: **Yes: **Verónica Díez Díaz

Reviewer #2: No

---

## [Editor Report · Acceptance letter]

20 Mar 2023

PONE-D-22-27354R1 

Back to life: Techniques for developing high-quality 3D reconstructions of plants and animals from digitized specimens 

Dear Dr. Clark:

I'm pleased to inform you that your manuscript has been deemed suitable for publication in PLOS ONE. Congratulations! Your manuscript is now with our production department. 

Kind regards, 

on behalf of

Dr. Judith Pardo-Pérez 

Academic Editor

PLOS ONE